# Attentional selection and communication through coherence: Scope and limitations

**Priscilla E. Greenwood[1], Lawrence M. Ward**[2]*

**1** Department of Mathematics, University of British Columbia, Vancouver, Canada, **2** Department of Psychology and Djavad Mowafaghian Centre for Brain Health, University of British Columbia, Vancouver, Canada

* lward@psych.ubc.ca

## Abstract

Synchronous neural oscillations are strongly associated with a variety of perceptual, cognitive, and behavioural processes. It has been proposed that the role of the synchronous oscillations in these processes is to facilitate information transmission between brain areas, the 'communication through coherence,' or CTC hypothesis. The details of how this mechanism would work, however, and its causal status, are still unclear. Here we investigate computationally a proposed mechanism for selective attention that directly implicates the CTC as causal. The mechanism involves alpha band (about 10 Hz) oscillations, originating in the pulvinar nucleus of the thalamus, being sent to communicating cortical areas, organizing gamma (about 40 Hz) oscillations there, and thus facilitating phase coherence and communication between them. This is proposed to happen contingent on control signals sent from higher-level cortical areas to the thalamic reticular nucleus, which controls the alpha oscillations sent to cortex by the pulvinar. We studied the scope of this mechanism in parameter space, and limitations implied by this scope, using a computational implementation of our conceptual model. Our results indicate that, although the CTC-based mechanism can account for some effects of top-down and bottom-up attentional selection, its limitations indicate that an alternative mechanism, in which oscillatory coherence is caused by communication between brain areas rather than being a causal factor for it, might operate in addition to, or even instead of, the CTC mechanism.

## Author summary

The ability to select some stimulus or stimulus location from all of those available to our senses is critical to our ability to navigate our complex biological and social niche, and to organize appropriate actions to accomplish our goals, namely to survive and to reproduce. We study here a possible mechanism by which the brain could accomplish attentional selection. We show that the interaction between 10 Hz and 40 Hz neural oscillations can provide increases in coherence and information transmission between computationally modelled cortical areas. Our results support the idea that synchronization between oscillations facilitates communication between cortical areas. Limitations and sensitivities of the model, however, also point to the idea that the relation between coherence and

**Data Availability Statement:** The simulation code used in this submission is available from https://github.com/lward77/CTC.git.

**Funding:** Preparation of this article was supported by a Discovery Grant, F19-05283, to LMW from

the Natural Sciences and Engineering Research
Council of Canada at https://www.nserc-crsng.gc.
ca/index_eng.asp. The funders had no role in study
design, data collection and analysis, decision to
publish, or preparation of the manuscript.

**Competing interests:** The authors have declared
that no competing interests exist.

communication could go in the opposite direction in some cases, particularly when selective attention is not involved.

## Introduction

Ever since the discovery of stimulus-specific, synchronized, neuronal oscillations, in both spiking and local field potentials (LFPs), in the gamma band (around 40 Hz) in the visual cortex of the cat [1], there has been speculation about the role of these oscillations in the brain. Of particular relevance to the present paper is the suggestion that information flow in the cortex could be facilitated by synchronization of the oscillations between different cortical areas [2–4]. This suggestion has been buttressed by the work of many researchers, e.g., [5–11] to cite only a few, who showed behaviourally-relevant synchronization in brain activity at various scales of time and space. The idea that synchronized neuronal oscillations could facilitate communication between various brain areas has come to be called 'communication through coherence' (hence CTC) [12, 13]. Many empirical studies have provided circumstantial evidence that CTC is indeed importantly involved in how information is transmitted between brain areas, e.g., [14–17] to cite only a few, but also c.f. [18, 19]. Thus, CTC as a theoretical idea has seemed to be on a relatively firm footing empirically, although definitive causal evidence is still lacking.

Importantly, though, CTC has been challenged as getting the causal order wrong (e.g., [18–20]). Instead, this work suggests, the order should be 'coherence through communication' (CTC′), or 'coherence resulting from communication.' In other words, neuronal coherence is said to be an epiphenomenon of communication between brain areas, rather than a facilitator of that communication. Most recently, Schneider and colleagues [20] have created a model, and run experiments, showing that simply having connections between two brain regions allows bursts of spikes in a sending area to create similar bursts in a receiving area, as well as coherence between the local field potentials (LFPs) in the beta band (around 20 Hz) in the two areas. Interestingly, in the experiments, the receiving area did not display significant oscillations in the beta band, even while displaying LFP coherence with the sending area in the beta band, nor did the receiving area display phase locking between spikes and oscillations. Notably, their Synaptic and Source-Mixing (SSM) model, which does a good job of explaining their empirical results, does not involve coupling of oscillators and does not involve phase-locking between the sender's and receiver's spiking activity. In the SSM model, coherence and Granger causality between the sending and receiving areas depends on connectivity (number of active synapses) between sender and receiver, power spectrum of the sender (frequency at which the spectrum power is relatively high), and the coherence between the LFP at the sender and the signal sent by the sender. Schneider and colleagues [20], in reviewing other relevant evidence, also suggested 'Likewise, changes in inter-areal coherence with cognition or behavior should be very carefully interpreted . . .' (p. 14).

In this paper we investigate the CTC concept in the context of a theoretical and computational model of a mechanism that could choose salient signals from among the many impinging on an organism. This model enables a convincing account of the mechanism, but also points to some important limitations to which it is subject. These limitations implicate the reverse model, CTC′, as possibly replacing, or operating along side, the original CTC model. One way in which both of the two mechanisms could be operating in the brain is as follows: the attentional selection mechanism operates using CTC to enhance information transmission for attended stimuli, whereas uninvolved brain areas typically operate as described by the

CTC′ concept, according to the SSM model of [20] or something like it, when communicating information that is not attended and/or does not reach consciousness. Indeed, the results of [20] were obtained in the absence of an attentional manipulation, so that this suggestion is at least plausible.

## Phase difference and attention

It is well known that voluntary orienting of attention to a specific location, or to a specific feature or object, enhances processing of that location or stimulus [21]. One way this might work is for there to be enhanced information transmission about the attended stimulus between the cortical areas that are processing the sensory information. In the light of the CTC hypothesis, it is then reasonable to suggest that such enhanced information transmission could result from synchronization of neural oscillations between the relevant cortical areas. A study by Saalman and colleagues [22] reported that oscillations in the alpha band (around 10 Hz), originating in the pulvinar nucleus of the thalamus, were closely tied to top-down attentional processing in the cortex. Moreover, a more recent study by Eredath and colleagues [23] demonstrated a causal role of pulvinar input in synchronizing neural activity between V4 and Lateral Intra-Parietal (LIP) cortex by reversibly inactivating the dorsal pulvinar.

There is, in fact, a large literature on the role of alpha frequency oscillations in cognitive processing, including in memory, attention, and other cognitive functions (e.g., [24–26]). Alpha oscillations have been argued to modulate large-scale communication in the brain [27, 28]. Motivated by the general importance of alpha oscillations in neural processing, and more specifically by the results of [22], Quax and colleagues [29] created a spiking neuron network model of two communicating cortical areas that received noisy sine-wave alpha input as if from the pulvinar nucleus. They showed that the alpha input organized the gamma oscillations in the cortical areas, increasing phase coherence, information transmission, and Granger causality between the areas, consistent with the CTC hypothesis. Surprisingly, the alpha inputs to the two cortical areas in their study had maximum effects for the specific phase difference of $\Delta\phi = -\pi/2$ between the 10 Hz oscillations sent to the two cortical areas.

Intrigued by this finding, Greenwood and Ward [30] wondered what was special about $-\pi/2$. They proved a lemma that showed that, in fact, phase coherence, defined as the phase locking index of [31], between summed sine waves, say $\alpha_1 + \gamma_1$ and $\alpha_2 + \gamma_2$, similar to those in the study of [29], was maximal when the phase difference in radians, $\Delta\phi$, between $\alpha_1$ and $\alpha_2$ equalled that between $\gamma_1$ and $\gamma_2$, i.e., in the case of the results of [29], $-\pi/2$. The finding of [29], that maximal effects of alpha inputs to gamma-oscillating cortical areas were found when $\Delta\phi = -\pi/2$, pointed to the possibility that there was a similar phase difference, $\Delta\psi$, between the gamma oscillations in the two modelled cortical areas. Greenwood and Ward then found that the results of [29] could be replicated using a firing-rate-based macro-model of cortical neural activity to generate the gamma oscillations while retaining the noisy sine wave input for the alpha oscillations. Their cortical rate model comprised a coupled Excitatory-Inhibitory, or EI, pair of linear stochastic differential Eqs (1), (2) and (3) (see Methods for the equations). This model is similar to a linearized version of the firing rate model originated by Wilson and Cowan [32]. The linear model of [30] produces quasi-cycle oscillations rather than the limit cycle oscillations produced by the original, nonlinear, Wilson-Cowan model. Quasi-cycles are noisy periodicities resulting from damped oscillations sustained by noise, e.g., [33]. In a subsequent part of their study, Greenwood and Ward implemented a similar rate model with EI pairs also generating the 10 Hz oscillations supposedly emanating from the pulvinar, replacing the noisy sine waves used by [29] with quasi-cycles. Their replication of the results of [29]

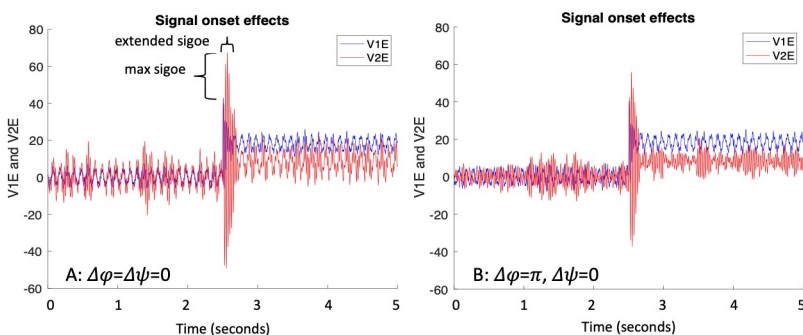

**Fig 1. Representative results of [30] (and the current study) demonstrating single realizations of gamma frequency 'cortical' quasi-cycle excitatory processes (V1E and V2E), with noisy alpha sine waves added to their respective, connected, inhibitory processes (V1I and V2I).** The V2E process is comprised of its own oscillations plus 0.5 times that of V1E with a lag of $\Delta\psi = 0$ radians. Signal = 20 onset occurs at $5 \times 10^4$ timepoints and continues until end of the run. The location and extent of maximum signal onset effect, 'max sigoe,' and mean signal onset effect, 'extended sigoe,' are indicated in the figure. See text for how the onset effects are computed. (A) Phase offset of the alpha waves is $\Delta\phi = 0$; max sigoe = 24.5, extended sigoe = 8.3, phase coherence between V1E and V2E $\rho = 0.64$. (B) Phase offset of the alpha waves is $\Delta\phi = \pi$; max sigoe = 15.35, extended sigoe = 3.76, $\rho = 0.51$.

using this method to produce oscillations completed the expression of a cortex-pulvinar model entirely in terms of quasi-cycles.

Fig 1 displays illustrative (both for [30] and the current work) single realizations of quasi-cycle oscillations for the excitatory processes in the two feed-forward-connected gamma-oscillating 'cortical' areas. These are the E processes of the EI pairs, where each E process is reciprocally connected to its respective I process (see Methods for equations). In the corresponding models, noisy alpha-frequency sine waves have been added to the increments of the inhibitory processes in each area, and are thus transmitted to the respective E processes through their E-to-I connection. The figure also illustrates the two main measures of signal transmission employed by [29, 30]. Maximum signal onset effect, 'max sigoe' in the figure, is defined as the difference between the amplitude of the positive part of the excitatory oscillation in the period from signal onset until 275 ms after signal onset in cortical area 2 minus that in cortical area 1. Extended signal onset effect, 'extended sigoe,' is defined as the mean positive excursion during the 50 ms on each side of the maximum in cortical area 2 minus that in cortical area 1. Apparent in the figure is, first, the transient burst of larger oscillations at signal onset, larger in V2E than in V1E, and, second, that the difference is larger, and the transient burst lasts longer, when the phase offset, $\Delta\phi$, of the alpha sine waves matches that of V1E and V2E, $\Delta\psi$. It is important to note that the burst of larger oscillations does not last for the entire duration of the signal; this feature will be discussed in the Results section. Also, one can see in the figure that the oscillations in V1E and V2E are more coherent with each other when the phase offsets match ($\rho = 0.64$) than when they do not ($\rho = 0.51$).

Fig 2 demonstrates the effects of alpha-gamma phase offset matching on phase coherence of the combined alpha-gamma oscillations, as well as on maximum and extended signal onset effects, over a range of phase offsets of the added alpha-frequency inputs to V1I and V2I, as in [29, 30]. Clearly the maximum of each of these measures occurs when the phase offset of the alpha waves approximately matches the phase offset of the gamma waves, near 0 in Fig 2. As mentioned, these results closely mirror those of [29], but with quasi-cycle oscillations rather than populations of spiking neuron models and sine waves. These results of [30] suggest that it is the oscillations that are essential in this situation, rather than the details of the neuron

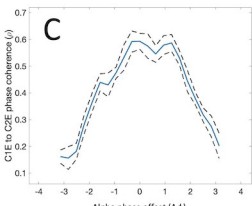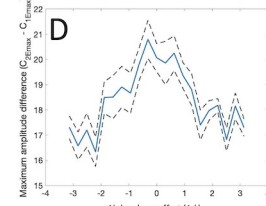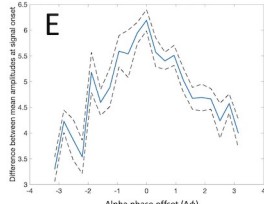

**Fig 2. Results of [30] demonstrating the effects of phase offsets of alpha (Δϕ) and gamma (Δψ) oscillations on (C) phase coherence of the combined oscillations, (D) maximum signal onset effect, and (E) mean signal onset effect.** Fig 1 displays representations of these effects. The *gamma* phase offset was Δψ = 0. Thus, the maximum phase coherence and signal onset effects occur near Δϕ = 0. Reproduced from Fig 4 of [30] with permission.

spiking that underlies the oscillations. The rate model in [30] has no spiking, only oscillations, but retains the results of [29].

Another recent study has also implicated phase differences between oscillations as important in information transfer between cortical areas [34]. In a context similar to that of [29], communicating populations of model neurons were simulated for two cortical areas. Here the frequencies of the oscillations in the sending and receiving areas, as well as the time delay (translated to a phase delay) between them, were varied. Again, in this study, an optimal phase difference was discovered; in this case information transmission and coherence were maximum at a phase delay of $\pi$ between the oscillations in the two areas. In the light of these additional findings, we undertook to elaborate the model of CTC studied in [29, 30], aiming to ascertain both its generality and usefulness as well as to reveal its limitations.

## Selective attention via phase offset

Here we describe an elaborated model based on that studied in [29, 30], where attentional selection is controlled by pulvinar input to communicating cortical areas (here assumed to be V1 and V2 in the visual system) as in [22, 35]. The model operates to focus attention on one out of several stimuli that are present at once, whether in a 'top-down,' voluntary, fashion or in a 'bottom-up,' involuntary, fashion, as attention does in the brain [21]. Fig 3 displays a conceptual picture of this model, which is an elaboration of Fig 2C in [30]. Within Fig 3 are two copies of Fig 2C in [30], with parts labelled X and Y. The two parts appear as separate if the inputs, signals X and Y, and the TRN are removed, although in actual cortex there might be connections between them. The two copies are representatives of the many 'channels', or subsystems, in each sensory cortical region, that are tuned to different features of stimulus inputs, including, in vision, e.g., position in space, colour, shape, and so forth. So, in Fig 3, channels X and Y represent groups of neurons in V1 that respond to the features of signals X and Y, respectively. The elaboration consists of the addition of circuitry involving the Thalamic Reticular Nucleus (TRN) and its interaction with the pulvinar generators of 10 Hz oscillations.

The elaborated system displayed in Fig 3 operates in top-down mode as described in [29, 30]. First, as in [29, 30], it is assumed, and implemented by parameter choices, that cortical areas oscillate at approximately 40 Hz, and pulvinar inputs to cortex oscillate at approximately 10 Hz. In the model of [30] both oscillations are produced internally to cortex and pulvinar as noisy quasi-cycles as defined in (1), although other generators of neural oscillations, such as PING models, are not being ruled out. Next, it is assumed that the phase offset, Δψ between V1 and V2 gamma oscillations, arising from the synaptic delay between V1 and V2 neurons, is fixed at the same value for all signals to which V1 neurons are tuned. The phase offset, Δϕ, of the 10 Hz pulvinar inputs to V1 and V2 is fixed at approximately the same value as the phase

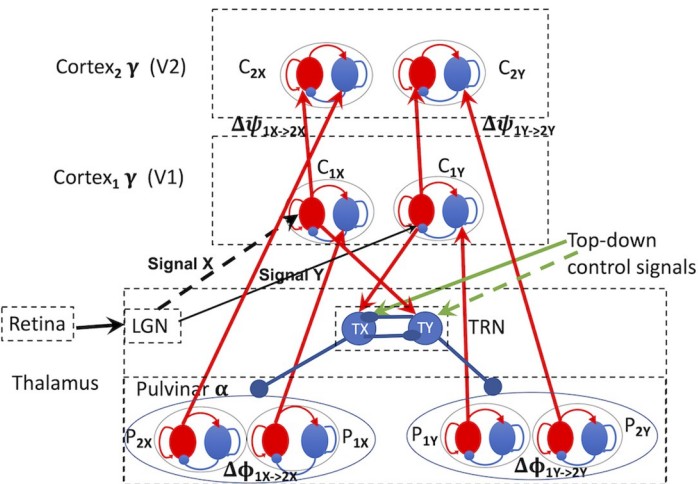

**Fig 3. Structural model of top-down and bottom-up attentional selection.** EI pairs are red (excitatory, E) and blue (inhibitory, I) disks; red arrows are excitatory, blue arrows are inhibitory, black arrows represent input signals, or stimuli,—they add increments in neural activity to the processes to which they connect. Top-down signals, green arrows, excite thalamic reticular nucleus (TRN) inhibitory neurons in the same way as do inputs from V1. The (visual) signals excite the V1 neurons that are tuned to them, e.g., vertical (say, signal X) or horizontal (say signal Y) lines on the retina excite V1 neurons tuned to vertical or horizontal lines respectively via tuned neurons in the LGN. See text for more detailed explanation.

offset between V1 and V2. Thus, the phase offsets between alpha and gamma oscillations approximately match, meaning, as shown in [29, 30], that there is relatively high coherence and information transmission between V1 and V2 when corresponding areas of V1 and V2 receive 10 Hz oscillations from the pulvinar. The TRN contains only inhibitory neurons and is connected to all of the 10 Hz oscillators in the pulvinar. Thus, the TRN can modulate alpha input to V1 and V2 by inhibiting the output of particular 10 Hz oscillators to V1 and V2. For example, in Fig 3 each of the two signals, X and Y, will excite their respective E processes of $C_{1X}$, $C_{1Y}$ in V1, which in turn excite both their corresponding E processes $C_{2X}$, $C_{2Y}$ in V2, and their opposite inhibitory processes in TRN, TY and TX. The inhibitory processes in TRN inhibit each other as well [36], so there is a competition between incoming stimuli implemented in the TRN. Exciting the inhibitory TRN processes will inhibit activity in the relevant pulvinar neurons, $P_{1Y}$ and $P_{2Y}$ for signal X and $P_{1X}$ and $P_{2X}$ for signal Y, to the extent permitted by the internal TRN competition and any top-down control signals. Note that when there are many objects in the visual field, as is usual, all of the areas of the TRN corresponding to those objects or locations will be inhibited except for the attended object or location.

In a typical top-down visual attention scenario, there would be a variety of stimuli available in the visual scene, say A,B,C,. . .X and Y. One of these, at some location and extent in the visual field, would be selected based on salience given by current goals, previous interactions, memory, etc. Top-down control signals implementing 'voluntary' attentional selection (re these goals, search targets, etc.) would affect the TRN neurons by strongly exciting those TRN neurons connected to cortical neurons tuned to stimuli *other than the desired stimulus*. In Fig 3 the to-be-attended stimulus initially is signal Y (via solid green arrow). Thus, the TRN would inhibit the 10 Hz pulvinar oscillations from going to those (unselected) tuned neurons in V1, $C_{1X}$, that are responding to signal X (and also any other stimuli in the visual field). This would allow the 10 Hz and 40 Hz phase-offset match to induce optimal coherence and signal transmission between $C_{1Y}$ and $C_{2Y}$ in V1 and V2 for the selected signal Y. If signal Y is selected in

this way, then phase coherence and signal transmission would remain suboptimal for signal X and any other signals because of lack of 10 Hz input, and signal Y would be 'attended.'

There is a nuance that should be mentioned here: when visual objects possessing a particular feature, e.g., colour red, are to be attended but are intermixed spatially with other objects with a different feature, say green, attention appears to operate by inhibiting the distractors rather than by exciting the targets [37]. The simple model of Fig 3 does not address this situation, although the basic elements of the inhibitory TRN, spatial maps of simple feature locations in cortex (e.g., a retinotopic map of colours—[38]), and connections from cortex to TRN are present. This would allow multiple locations to be excited in TRN, thus inhibiting multiple locations in pulvinar, and reducing input levels for the inhibited locations. If the features are not simple and represented in multiple locations in a retinotopic map, however, but rather are complex parts of objects distributed over spatial locations, a more sophisticated version of our system must be called upon. Presumably the relevant representations of these features are located in higher areas of the visual system, and thus attending to them could be a somewhat different process, although still possibly mediated by alpha-frequency oscillations [28].

Competition between TRN neurons would mediate between signals of different strengths and top-down selected signals for precedence, as well as between signals in different sensory modalities (and primary sensory cortical areas). Top-down attention can overcome some differences in signal strength as depicted in our example (signal X arrow is thicker, e.g., [39]). Some signals, however, are simply too strong, e.g., a very bright flash of light. Let us assume that signal X suddenly flashes brightly, indicated by the X line being dashed. In this case the the bottom-up, 'involuntary,' or 'automatic,' mechanism might pull attention away from whatever top-down signal was being selected at the time [21]. We assume that, as in behavioural studies [21], this pulling of attention to a new stimulus typically is followed by reorienting top-down attention to the new stimulus until its relevance is ascertained. This is not a necessary action, however, and bottom-up attentional enhancement effects are typically observed substantially earlier than in the case of top-down orienting, around 100-200 ms after such a flash, indicating that the orientation of top-down attention is not required to produce the attention effect. Even though attention currently is being paid to signal Y through the top-down mechanism just described, the brightening of signal X will have several effects. First, it will produce a very strong signal onset effect, in spite of the fact that no pulvinar 10 Hz oscillation is being sent there because of the allocation of top-down attention to signal Y (see *Results* for a supporting simulation result). This will be transmitted to executive areas of the brain responsible for sending signals to implement top-down attention. Also, because the very strong signal X wins the competition between the TRN neurons, and this will be reinforced after some time by a top-down signal exciting TY (indicated by the dashed green arrow), 10 Hz input from $P_{1Y}$ and $P_{2Y}$ to $C_1Y$ and $C_2Y$ will be terminated. Thus, 10 Hz oscillations from $P_{1X}$ and $P_{2X}$ now will be injected into the 40 Hz oscillating $C_1X$ and $C_2X$ in V1 and V2, respectively. The roughly equal phase offsets between 10 Hz and 40 Hz oscillations for signal X now will optimize the phase coherence and signal transmission between V1 and V2 for signal X, as described in [30]. Because the 10 Hz input from $P_{1Y}$ and $P_{2Y}$ to $C_1Y$ and $C_2Y$ has been blocked by TY, however, the coherence and signal transmission between relevant cortical areas become lower (but not 0) for signal Y than for signal X. The large signal onset effect of signal X at V2 (and likely further up the visual system), is presumed to be what it means for signal X, the very strong signal, to 'pull,' or 'draw,' attention to itself in a bottom-up manner. There is a wealth of evidence that this effect is ubiquitous, both in the brain and in behaviour [21].

Our contribution here is the above description of a possible mechanism for the interaction between the pulvinar, the TRN, and the cortex, to select an object of attention. The question

arises: to what extent is this mechanism dependant upon oscillations? Our conception of an implementation is in terms of quasi-cycles, which we now introduce.

## Methods

### Computational implementation of the model

As in [30], here we use an EI rate model that produces quasi-cycle oscillators that are described in detail in [40] based on a model of [41]:

$$d\mathbb{V} = -\mathbb{A}\mathbb{V}dt + \mathbb{N}d\mathbb{W} \tag{1}$$

where

$$\mathbb{V} = \begin{pmatrix} V_E(t) \\ V_I(t) \end{pmatrix}, \quad \mathbb{A} = \begin{pmatrix} (1 - S_{EE})/\tau_E & S_{EI}/\tau_E \\ -S_{IE}/\tau_I & (1 + S_{II})/\tau_I \end{pmatrix} \tag{2}$$

$$\mathbb{N} = \begin{pmatrix} \sigma_E & 0 \\ 0 & \sigma_I \end{pmatrix}, \quad d\mathbb{W} = \begin{pmatrix} dW_E(t) \\ dW_I(t). \end{pmatrix} \tag{3}$$

This model is a description of the time evolution of the local field potentials (LFPs), closely related to firing rate, of two coupled populations of neurons, excitatory, $V_E(t)$, and inhibitory, $V_I(t)$. The $V_E$ and $V_I$ processes defined by (1), (2) and (3) will be called the E and I processes. We refer to a stochastic process defined by a model of the form (1) as an EI pair. The parameters in $\mathbb{A}$ represent the synaptic coupling strengths, $S_{ij}$, and time constants, $\tau_i$, between the excitatory and inhibitory neuron populations, $\sigma_i$ are noise amplitudes, and $W_i$ are standard Wiener noise processes. In this model, when the eigenvalues of $-\mathbb{A}$ are complex, $-\lambda \pm i\omega$, with $0 < \lambda \ll \omega$, the system (1) has sustained (noisy) oscillations, called quasi-cycles, distributed in a narrow band around $\omega$. The parameters in $-\mathbb{A}$ can be adjusted to give a wide range of quasi-oscillation frequencies within the range relevant to brain activity related to mental processes [40]. The parameter values used in $\mathbb{A}$ to produce approximately 40 Hz oscillations in the EI pairs are listed in Table 1. The left hand side of the diagram in Fig 3, without the TRN or signals (or, similarly, the right hand side by itself), was the subject of [30]. Simulations in [30] used (1) to produce independent quasicycles of central frequency gamma in the EI pairs denoted $C_{1X}$, $C_{2X}$, and of central frequency alpha in the EI pairs denoted $P_{1X}$, $P_{2X}$ in Fig 3.

In this paper we have opted to implement simulations of the components of the elaborated model in the simplest possible way, given that their main effects have already been demonstrated in both populations of individual spiking neuron models [29] and of generic rate models like (1) [30]. Both types of models emphasize periodic alpha and gamma oscillations, although the population model of [29] exhibits the oscillations in bursts of spiking organized by a sinusoidal input, whereas the rate model studied in [30] and here exhibits interacting

**Table 1. Parameters of $\mathbb{A}$ used in 40 Hz EI pair simulations.**

| Variable | Value | Units |
|:---:|:---:|:---:|
| $S_{II}$ | 0.1 | dimensionless |
| $S_{EE}$ | 1.4 | dimensionless |
| $S_{EI}$ | 0.78 | dimensionless |
| $S_{IE}$ | 2.1105 | dimensionless |
| $\tau_E$ | 0.003 | seconds |
| $\tau_I$ | 0.006 | seconds |

quasi-cycles. The process defined by (1) can be factored into a rotation (sinusoid) multiplied by a 2D Ornstein-Uhlenbeck process [33], and so is essentially a noisy sinusoid [33].

Each of the two signal channels in the model of Fig 3 is computed as follows. The channels of the cortical processes (V1 and V2) are modelled as

$$d\mathbb{C}_1 = [-\mathbb{A}_C\mathbb{C}_1 + b_1\mathbb{B}_1]dt + signal + s_1 d\mathbb{W}_1 \tag{4}$$

$$d\mathbb{C}_2 = [-\mathbb{A}_C\mathbb{C}_2 + c_{21}\mathbb{C}_{21}(\Delta\psi) + b_2\mathbb{B}_2(\Delta\phi)]dt + s_2 d\mathbb{W}_2, \tag{5}$$

where $\mathbb{A}_C$ is the $\mathbb{A}$ matrix of coefficients that generates 40 Hz oscillations, damped but sustained by noise, $\mathbb{C}_1, \mathbb{C}_2$ are 2D vector processes, $(C_{1E}, C_{1I})'$, $(C_{2E}, C_{2I})'$, for EI pairs, as in (1), (2), (3), oscillating at natural frequency 40 Hz but with $\mathbb{B}_1 = (0, -\alpha_1(t))'$, $\mathbb{B}_2(\Delta\phi) = (0, -\alpha_2(\Delta\phi, t))'$, added incrementally, and where $b_1$, $b_2$ are the amplitudes of the alpha sine waves injected into $C_{1I}, C_{2I}$. Further, $\mathbb{C}_{21}(\Delta\psi) = (C_{1E}(t + \Delta\psi), 0)'$ represents the input to $C_{2E}$ from $C_{1E}$ delayed by $\Delta\psi$, where $\Delta\psi$ is the phase difference between $C_{1E}, C_{2E}$ caused by adding a fraction, $c_{21}$, of $C_{1E}$ to $C_{2E}$ at a time delay that implements the phase offset between V1 and V2. The *signal* in (4) is modelled as a constant input, i.e., a step function, as in [29, 30], except where indicated. Although common to both in vivo and in vitro studies, this way of introducing a signal to the model is problematic because LGN input to V1 actually consists of structured bursts of spikes [42]. This point will be discussed in *Sensitivities and Limitations*.

As we demonstrated in [30], modelling the pulvinar 10 Hz inputs as noisy sine waves or as EI pair quasi-cycles gives rise to similar results. Here we use noisy sine waves for simplicity and to enable the most precise phase offset of the alpha inputs. The inputs from the noisy sine waves oscillating at approximately 10 Hz are:

$$\alpha_1(t) = M_\alpha \sin(2\pi 10 t) + s_\alpha S_{\alpha 1}(t) \tag{6}$$

$$\alpha_2(\Delta\phi, t) = M_\alpha \sin(2\pi 10 t + \Delta\phi) + s_\alpha S_{\alpha 2}(t), \tag{7}$$

where $M_\alpha$ is the amplitude of the alpha sine waves and $\Delta\phi$ is the $\alpha_1$ to $\alpha_2$ phase difference. The white noise, $S_\alpha(t)$, added to the alpha sine waves has standard deviation $s_\alpha$.

Sample paths of processes $\mathbb{C}_1, \mathbb{C}_2$, defined by Eqs (4) and (5) were simulated using the Euler-Maruyama algorithm, with the current values of the noisy 10 Hz sine waves added to the rhs of the SDE at each time point.

Table 2, which is the main focus of this paper, displays the results of a large number of simulations with strategically chosen representative values for the parameters of Eqs (4)–(7). The simulations were run for 100,000 time points at 0.00005 sec per time point, so for 5 sec. A signal was added at time point 50,000 and remained on until time point 100,000. As in [30], phase coherence, $\rho$, was calculated as $\rho = \frac{1}{N}\left|\sum_{j=1}^{N}\exp(i(\theta_{V1,j} - \theta_{V2,j}))\right|$, [31], where $\theta$ is the instantaneous phase of the V1 or V2 oscillation computed from the analytic signal via the Hilbert transform, and $N$ = number of time points over which $\rho$ is computed. Here $N = 100,000$ time points. Signal onset effects, as in [29, 30], were measured as (1) the maximum (max) of the positive part of the oscillation in V2 minus that in V1 between time points 50,000 and 55,000 (250 ms after signal onset), and (2) the mean positive part of the oscillation over the interval max-1000 and max+1000 timepoints (50 ms).

Mutual information, *MI*, was computed between V1 and V2 using a Matlab function [43] as

$$MI(V2; V1) = H(V2) - H(V2|V1), \tag{8}$$

**Table 2. Results of simulations of model in Fig 1 using selected values of the parameters appearing in Eqs (4)–(7).** Parameter values ($\Delta\phi$, $\Delta\psi$ in radians) are in the left-most seven columns, output values in the rightmost four columns. "sig noise" is signal noise; "max sigoe," the maximum signal onset effect, is the maximum of the positive part of the oscillation at V2 minus that at V1, i.e., $max(V_{2+} - V_{1+})$, between signal onset and signal onset + 250 ms; 'extended sigoe,' the extended signal onset effect, is the mean of the positive part of the oscillation between time at maximum −50 ms and time at maximum + 50 ms at V2 minus that at V1; MI is mutual information between the phases of V1 and V2 after signal onset until the end of the run. All output values in the table, $\rho$, max sigoe, extended sigoe, and MI, are the means of 10 realizations, with standard errors approximately 0.01, 0.95, 0.32, and 0.02 resp.

| Row | $M_\alpha$ | $\Delta\phi$ | $\Delta\psi$ | $s_1 = s_2$ | $c_{21}$ | signal | sig noise | $\rho$ | max sigoe | extended sigoe | MI |
|---|---|---|---|---|---|---|---|---|---|---|---|
| 1 | 6 | 0 | 0 | 0.03 | 0.5 | 20 | 0 | 0.69 | 17.6 | 6.5 | 1.54 |
| 2 | 0 | na | 0 | 0.03 | 0.5 | 10 | 0 | 0.29 | 8.4 | 2.6 | 1.07 |
| 3 | 0 | na | 0 | 0.03 | 0.5 | 20 | 0 | 0.5 | 15.6 | 4.7 | 1.34 |
| 4 | 0 | na | 0 | 0.03 | 0.5 | 30 | 0 | 0.73 | 22.3 | 5.08 | 1.59 |
| 5 | 6 | $\pi$ | 0 | 0.03 | 0.5 | 20 | 0 | 0.45 | 18 | 4.48 | 1.27 |
| 6 | 6 | 0 | 0 | 0.03 | 0.1 | 20 | 0 | 0.44 | -27.6 | -8.0 | 1.58 |
| 7 | 2 | 0 | 0 | 0.03 | 0.5 | 20 | 0 | 0.53 | 17.7 | 5.3 | 1.33 |
| 8 | 6 | 0 | 0 | 0.03 | 0.8 | 20 | 0 | 0.71 | 54.2 | 17.4 | 1.52 |
| 9 | 2 | 0 | 0 | 0.03 | 0.8 | 20 | 0 | 0.56 | 51.7 | 15 | 1.37 |
| 10 | 0 | na | 0 | 0.03 | 0.8 | 20 | 0 | 0.52 | 52 | 14.65 | 1.39 |
| 11 | 10 | 0 | 0 | 0.03 | 0.8 | 20 | 0 | 0.66 | 54 | 16.5 | 1.62 |
| 12 | 6 | 0 | -2 | 0.3 | 0.5 | 20 | 0 | 0.5 | 55.1 | 13.3 | 0.92 |
| 13 | 0 | na | -2 | 0.3 | 0.8 | 20 | 0 | 0.34 | 126 | 38 | 1.05 |
| 14 | 6 | 0 | 0 | 0.03 | 0.5 | 20 | 0.5 | 0.7 | 18.3 | 6.6 | 1.55 |
| 15 | 0 | na | 0 | 0.03 | 0.5 | 20 | 0.5 | 0.66 | 15.9 | 4.5 | 1.43 |
| 16 | 16 | 0 | 0 | 0.03 | 0 | na | na | 0.53 | na | na | 2.2 |
| 17 | 0 | na | 0 | 0.03 | 0.5 | 20 | 0 | 0.75 | -40.6 | -9.1 | 2.9 |
| 18 | 6 | 0 | 0 | 0.03 | 0.5 | 20 | 0 | 0.72 | -37.4 | -8.4 | 2.5 |

where $H(V2)$ is the entropy of V2, and $H(V2|V1)$ is the entropy of V2 conditional on V1. MI was computed separately for the interval before signal onset and for the interval after signal onset, for the raw time series of V1 and V2, and also for their phases and amplitudes. Phases and amplitudes were computed from the analytic signal and the Hilbert transformed time series. As the results for the various MI computations were similar, only those for phase-based MI for the interval after the signal onset are reported here.

## Results

### Attentional selection mechanism

The first seven rows of Table 2 further develop the results of [30], and illustrate, for selected parameter values, how the model of Fig 3 works for the bottom-up and top-down attentional selection scenarios. Row 1 is an example where a strong signal, $X = 20$, wins the competition in the TRN and thus prevents $\alpha$ oscillations from the pulvinar from reaching V1 and V2 neurons tuned to the weaker signal, $Y = 10$. For the stronger signal, $\alpha$ oscillations at the matching phase offset, 0 in this case, would reach the areas of V1 and V2 that respond to that signal, causing a fairly large phase coherence value mean $\rho = 0.69$ (mean over 10 realizations), a strong signal onset enhancement of 'max sigoe' = 17.6, and also a strong extended signal enhancement effect of 'extended sigoe' = 6.5. These values are consistent with those for the matching phase offset case in [30] (see Fig 1 for an illustrative single realization and Fig 2 for average results over several different phase offsets).

For the weaker signal, Y, however, no $\alpha$ oscillations from the pulvinar reach V1 or V2 because of inhibition from the TRN. This situation is portrayed in Row 2 of Table 2. There, all

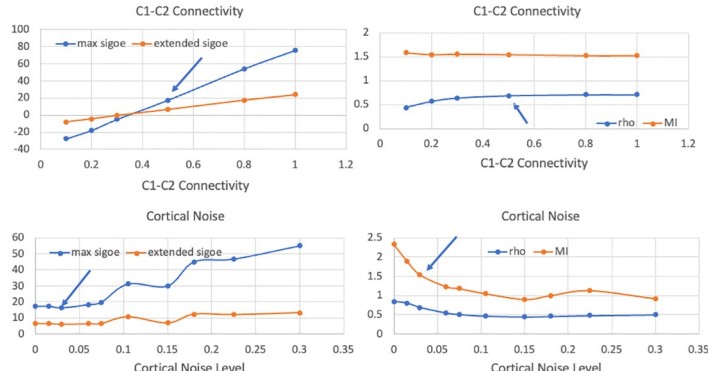

**Fig 4. Results of simulations of model in Fig 3 using selected values of the parameters appearing in Eqs (4)–(7).** The graphs depict the effects of cortical connectivity and cortical noise on the four outcome variables. Each point represents the average of 10 realizations. The standard errors are approximately the size of the data points or smaller and thus are not displayed. Parameter values other than the one changed are those in Row 1 of Table 2. The arrow points to an outcome listed in Table 2 for the indicated value of the changing parameter. MI = mutual information.

but the signal strength being the same, V1-V2 phase coherence, $\rho$ is substantially lower, as are the maximum and extended signal enhancement effects, consistent with both [29] and with [30]. Mutual information, 'MI,' between V1 and V2 during the signal period is also lower when there are no $\alpha$ oscillations sent to V1 and V2 for signal Y. Making $\Delta\phi = \pi \neq \Delta\psi = 0$, on the other hand, even with the fairly strong signal X = 20, reduces both $\rho$ and MI and the extended signal enhancement effect (Row 5), similar to results of [29, 30] (see Figs 1 and 2). Note that reducing the *magnitude*, $M_\alpha$, of the $\alpha$ oscillations to 2 from 6, as in Row 7, has only a small effect on $\rho$ and mutual information, and no effect on signal enhancement relative to Row 1 values. So at least some input from the pulvinar would seem to be sufficient for the attentional effects we see.

Row 3, compared with Row 1 of Table 2, shows the effects of top-down attention selecting one of two signals. If the two signals have equal magnitude, then the selected one (Row 1) still shows greater phase coherence, signal onset effects, and mutual information than does the unselected one (Row 3), again because the TRN has been signalled to inhibit $\alpha$ coming from the area of the pulvinar connecting to the unselected neurons in V1 and V2.

On the other hand, if a very strong signal, say X = 30, suddenly appears, attention can be pulled away from its current focus by very large signal onset effects, as in Row 4, despite the fact that, at the moment the strong signal appears, no pulvinar 10 Hz oscillations are going to the neurons tuned to the strong signal. Subsequently, however, 10 Hz oscillations would be sent to those neurons because of the mutual effects of the strong signal and top-down signal exciting TY, thus inhibiting $P_{1Y}$, $P_{2Y}$, and releasing $P_{1X}$, $P_{2X}$ from inhibition by TX.

Thus, our elaborated model uses the same basic mechanism, TRN inhibition of pulvinar $\alpha$ oscillations, to accomplish both top-down and bottom up attentional selection. In the next section we will see that the results displayed in Table 2, Rows 6-18, and those in Figs 4–6, point to a number of important sensitivities and limitations of this model.

## Sensitivities and limitations

In evaluating a model such as the one studied here, one is interested in its behaviour not only at the optimal parameter values, but also elsewhere in the parameter space. A useful model will be sensitive to parameter changes in a way that mirrors the behaviour of the phenomenon

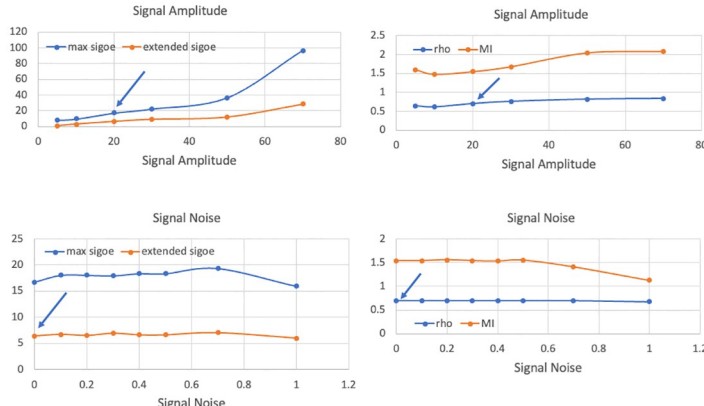

**Fig 5. Results of simulations of model in Fig 3 using selected values of the parameters appearing in Eqs (4)—(7).** The graphs depict the effects of signal amplitude and signal noise on the four outcome variables. Each point represents the average of 10 realizations. The standard errors are approximately the size of the data points or smaller and thus are not displayed. Parameter values other than the one changed are those in Row 1 of Table 2. The arrow points to an outcome listed in Table 2 for the indicated value of the changing parameter. MI = mutual information.

itself, while not oversensitive to small changes to which the phenomenon would be robust. In our present case these questions present difficult challenges. To begin to meet these challenges and further explore the parameter space of the model studied here, we extended the simulations illustrated in rows 6-18 of Table 2 and in Figs 4–6. Not surprisingly, the "success" of any model depends on the balance of several effects. When balance fails, pathologies are known to arise. Some of the sensitivities we are about to discuss could point to a source of pathologies in attention, or in neural transmission more generally, or they could simply arise from limitations of the model.

As we have shown, the elaborated model of selective attention can account for some basic attentional phenomena, in particular the association of phase coherence with information transmission between neural areas, as in the CTC hypothesis. There are, however, important limitations to the model, which we will illustrate using Table 2 and Figs 4–6. Some are obvious:

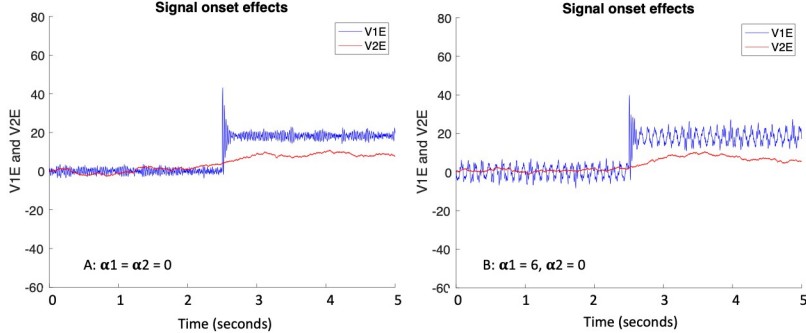

**Fig 6. Results of single realizations of simulations with gamma oscillations only in the V1 process.** The V2E process is comprised of random noise plus 0.5 times that of V1E with a lag of $\Delta\psi = 0$ radians (9), whereas the V2I process is comprised of random noise only (10). Note that V2E and V2I are not connected in these simulations. Signal = 20 onset occurs at $5 \times 10^4$ timepoints and continues until end of the run. (A) No alpha input to V1 or to V2 processes: max sigoe = -36.8, extended sigoe = -5.7, phase coherence between V1E and V2E $\rho = 0.8$, MI = 2.8. (B) Alpha input only to V1I process: max sigoe = -35.4, extended sigoe = -7.3, $\rho = 0.76$, MI = 1.9.

we are modelling at a macro scale, our EI equations are rate approximations, parameter values are not derived from empirical data, the details of both cortical and thalamic (TRN and pulvinar) connectivity are missing, and so forth. Importantly, it seems we and others (e.g., [29]) have found a sweet spot in parameter space within which the indicated mechanism operates. But parameter values outside that sweet spot yield a somewhat different picture. For example, the connectivity between V1 and V2 is critical. Reducing the connectivity from 0.5 to 0.1, as in Row 6 of Table 2 and Fig 4, yields moderate phase coherence and unchanged mutual information, but *negative* signal onset enhancement. Increasing the V1-V2 connectivity while leaving everything else unchanged, as in Row 8 and Fig 4, leads to values of $\rho$ and MI comparable to those for Row 1, but accompanied by much larger values for signal enhancement. Finally, increasing the amplitude of the $\gamma$ quasi-cycles by a factor of 10, to 0.3, by increasing the noise in the SDEs for V1E and V2E, as in Row 12 and Fig 4, leads to a relatively small value for $\rho$ and MI, but a very large value for signal enhancement. Abolishing the $\alpha$ input under those conditions, Row 13, gives rise to a large value of signal enhancement but still accompanied by relatively small $\rho$ and MI. In both of these latter cases, the centroid of the distribution of phase differences between V1 and V2 was shifted by the noise to -2 radians, not matched to the 0 phase offset of the $\alpha$ frequency. These values are clearly inconsistent with CTC. They are also inconsistent with the idea that phase offset-matching of 10 Hz input to cortex is the most important factor in affecting signal transmission between cortical areas.

Another aspect needing attention in both our and other models is the question of how input signals (stimuli) should be modelled. Some authors, e.g., [29, 30], have used a constant, i.e., step function, signal. Others have used a burst of Poisson or Gaussian pulses (basically, noise), and still others have used a constant plus some noise. Some, e.g., [44], have used a signal modelled on actual output from the LGN as input to V1. It does appear, however, that using as input the output from a noisy integrate and fire neuron (with a constant input to the model neuron) provides a good approximation of LGN output statistics, and has been shown to give better orientation selectivity than does LGN output modelled as an inhomogeneous Poisson point process [42]. Use of a constant as our input signal represents another limitation of our and others' model results. In most of our simulations we have used a step function as a signal, as in [29]. When we increase the amplitude of this step function, as in Fig 5, we find an increase in the signal onset effects as well as in phase coherence, $\rho$, and in mutual information. When we input a noisy signal, defined as a constant plus Gaussian noise (called 'signal noise'), to our EI modelled V1, as in Row 14 of Table 2 and Fig 5, we see little difference in signal onset effects, $\rho$, or MI compared to those in Row 1, but a decrease in signal onset effects and mutual information when the signal noise reaches higher levels. Without $\alpha$ input however, Row 15, these effects are only slightly reduced. So noise, both cortical and in the signal, if present, is apparently playing a role in the signal enhancement effect. Note that both our quasi-cycle model, here and in [30], and the neuron population model of [29], involved noisy (stochastic) processes.

**Coherence with no connection.**   The CTC hypothesis implies a direct connection between two cortical areas, because information is being transmitted from one to the other. In CTC, the spikes from the sending area must arrive at the receiving area at an optimal point in the phase resetting curve. In addition, the CTC′ hypothesis also requires a direct connection between the sending and receiving areas. But such a connection is not necessary for there to be phase coherence between two cortical areas. For example, Row 16 of Table 2 shows that when the V1-V2 connection is zero, but the $\alpha$ driving is large, high phase coherence is observed between V1 and V2, along with even higher MI. Of course, with no direct connection there is no direct information transmission between the areas. Such a result can mislead analysis of neural data for functional connectivity when several sources are simultaneously active but only

simple pairwise phase coherence is measured. In these cases significant coherence between two sources could arise because each is coherent with a third source, or even multiple other sources. Most authors are aware of this problem, but more appropriate multivariate analysis is more complicated, and so is seldom done.

**Coherence through communication.** It is clear from the results displayed in Rows 8-11 in Table 2 that driving of one cortical area by another can result in high phase coherence and MI, as well as large signal onset enhancement effects, regardless of the magnitude (or phase) of $\alpha$ input to them, including none at all (Row 10). Rows 17 and 18 of Table 2 display results of simulations we performed to test this idea in the context of our model. In these simulations we had V1E and V1I oscillating as in (4), but changed the V2E and V2I processes as follows:

$$dV2E = [-V2E + c_{21}V1E]dt + s_2 dW_E, \tag{9}$$

$$dV2I = s_2 dW_I. \tag{10}$$

These processes display no oscillations natively, with independent noises, and are not connected to each other. Process V2E is connected to V1E by the parameter $c_{21}$, as in previous simulations, which adds this proportion of V1E activity to that of V2E. Fig 6 shows single realizations of V1E and V2E (A) with no alpha input to V1E, V1I or V2E, V2I, and (B) with alpha input only added to V1I, and thus to V1E as the two are connected, as in (4).

There are several notable aspects of the results in Rows 17 and 18 of Table 2 and Fig 6. First, Fig 6 shows that the oscillations in V1E do not induce oscillations in V2E. Spectral power in the gamma band is present as usual in V1E, and also appears in the alpha band when the noisy alpha sine wave is added to V1I. However, there is no spectral power whatsoever in the gamma or in the alpha band in V2E. Second, note in Fig 6 that the average level of V2E rises from around 0 to around 10 after the signal appears; it remains at around 0 if no signal is present, as in the pre-signal period. Therefore, the signal level is transmitted from V1E to V2E but no oscillation in V2E is needed for this to happen. A detector in the visual system should be able to detect the presence of the signal in the change of the firing rate of V2E. Third, although there is a burst of increased oscillations at signal onset in V1E, Rows 17 and 18 of Table 2 show that signal onset effects are missing. Actually they are negative because of the lack of a burst of increased (oscillation) amplitude in V2E because of the absence of oscillations there. Moreover, phase coherence and MI are actually larger in this scenario than in the previous simulations, which had native oscillations in V2E and V2I.

This is somewhat similar to the scenario envisioned by the CTC' hypothesis espoused by [20], and also similar to that studied by [34], except in the latter study the second cortical area also displayed oscillations. As mentioned in the introduction, it is possible that this mechanism could be responsible for coherence between cortical areas when information is being transmitted outside of attention or consciousness, even while the pulvinar-origin $\alpha$ mechanism could be responsible for attentional enhancement of sensory inputs via the signal onset effects. Alternatively, it is possible that pulvinar-origin $\alpha$ input to cortex has some other function, somehow associated with attention but not actually causing greater phase coherence and enhanced information transmission.

**Onset vs total signal effects.** In the models studied by [29, 30] and here, the largest effects of $\alpha$ input to modelled cortical areas were at signal onset. As shown in previous sections, this is affected by phase offset matching but also by connectivity between areas (at least the way we compute it, as incrementally additive input) and by noise. There seems to be a smaller effect on ongoing information transmission, at least as computed by MI over the entire time the signal is present. (We note that it is necessary to have a relatively long time series to compute MI

in an unbiased way, as is also the case for phase coherence). It can be seen in Fig 1, however, that the value around which the post-signal process oscillates in V2E continues to be affected by the input from V1E, which includes the constant signal times the connectivity, which was 0.5 in Fig 1. There the signal to V1E was 20, and so the transmitted value to V2E was around 10, which is the value around which the V2E oscillations occur after the onset effect dissipates. Thus, the signal effect does persist as long as the signal persists, although not as dramatically as at signal onset. The smaller effect on ongoing signal transmission is possibly advantageous in the context of a complex environment, allowing for other signals to interrupt with their own large onset effects or change in coherence/signal transmission.

## Discussion: On the role of alpha oscillations

As briefly reviewed in the Introduction, there are quite a few studies indicating an important role for phase-synchronized gamma-frequency oscillations in information transmission in the cortex. In contrast, there are very few studies describing a mechanism by which such synchronization could be achieved. The experiments by Saalman et al. [22] and Eredath et al. [23] suggested that alpha band oscillations originating in the pulvinar nucleus of the thalamus could provide such a mechanism, and the modelling studies of [29, 30], and including the present work, explored such a mechanism computationally. To our knowledge this comprises the list of such studies. In spite of other work indicating a range of roles for alpha-frequency oscillations in the brain, e.g., [25, 26, 28], a number of issues remain where the mechanism we have studied is concerned. Here we discuss these issues in the context of the present work.

First, let us consider what would be required as a rigorous test of what we will call 'the alpha-gamma model,' the model in which alpha oscillations organize gamma oscillations [29, 30] as elaborated in the present study. We would need to see an experimental observation in the brain of the same kinds of changes in alpha and gamma power and phase synchronization, including gamma power (or oscillation amplitude) locking to alpha phase, at the indicated moments after an attention cue or salient stimulus, as those predicted by [29, 30] (see their Figs 3 and 4, 3 and 4, resp.), and by the present elaborated model. Moreover, we would need to record neural activity from, or intervene at, both relevant subcortical (e.g., pulvinar) and cortical sites. Interestingly the earlier models described in [29, 30], and the present, elaborated, model, make slightly different predictions about such an experiment.

Consider an experiment using a monkey preparation in which a (visual) cue as to where in space to attend is presented, followed at some time interval by a visual stimulus to be responded to, either presented at the attended-to location or at an unattended location. In such an experiment recordings of neural activity would be made from relevant cortical areas in, say, visual cortex, e.g., V1 and V4, that have specific receptive fields on the retina or in other areas, such as the Frontal Eye Fields (FEFs), that are involved in visual attention, as well as from pulvinar areas projecting to the same cortical areas. If the stimulus is presented to the attended location, then the receptive field relevant to that location is an 'attend-in' receptive field; if the stimulus is presented to the unattended location then the receptive field for that location is an 'attend-out' receptive field. A typical result in such experiments, e.g., [45–48], is that gamma-frequency phase coherence, or overall coherency, between and within relevant cortical areas, is higher in attend-in than in attend-out situations in a specific time frame after the stimulus presentation. This result is, of course, consistent with predictions from CTC, the alpha-gamma hypothesis, and either of the earlier [29, 30] or present models. It does not, however, really address the role of alpha-band oscillations in the synchronized gamma-band activity, or the timing of various changes in alpha and gamma power and phase coherence from the

moment of presentation of the attention cue until some time after the stimulus has been presented.

The various models diverge with respect to their predictions about alpha coherence and power in the two cortical areas, as well as those effects in subcortical regions. CTC is silent as to power or coherence specifically in the alpha (or lower) frequency range for both cortex and subcortex, although it could be presumed that a simple reading of CTC would predict that phase coherence at any frequency would facilitate neural communication at that frequency. Interestingly, some cortical studies that address the CTC hypothesis do present results about low frequency coherence and power, albeit as a side finding to the major focus on gamma activity. For example, Fries et al. [45] showed that alpha and gamma coherency (includes both amplitude and phase coherence) of both spiking and Local Field Potentials (LFPs) in monkey V4 are in opposition: gamma coherency is high in attend-in and low in attend-out conditions, and vice versa for alpha coherency. Bosman et al. [46] reported that gamma (60-80 Hz) phase LFP coherence and Ganger causality between monkey V1 and V4 was significantly higher during attend-in conditions but that this difference was not seen in lower frequencies. Nonetheless, lower-frequency coherence and Granger causality were relatively high in both attend-in and attend-out conditions even though lower-frequency power was low throughout. Grothe et al. [47] reported a very similar result in a similar preparation, especially the lack of significant attentional modulation of coherence between monkey V1 and V4 in the alpha/beta frequency band even though there was significant phase coherence in that band. Finally, Gregoriou et al. [48] showed that a variety of measures of coherence as well as Granger causality between monkey Frontal Eye Fields (FEF) and V4 are greater for attend-in conditions in the 40-60 Hz range, whereas these measures did not differ significantly in the same way with attention conditions in the 10 Hz range, even while coherence was non-zero in that range. These results for alpha frequencies are more complicated than the CTC hypothesis would suggest, and could be construed to be inconsistent with the alpha-gamma hypothesis and with the present model. As will be seen, however, it is not easy to predict alpha power or coherence results from any of the models.

The problem with prediction about alpha frequency observations in cortex alone in the suggested experiment stems from a variety of other experimental and model findings. As we and Quax and colleagues have shown, alpha power should *increase*, relative to some baseline, in modelled cortical areas when they receive 10 Hz input from the pulvinar [29, 30]. However, many experimental studies have shown that alpha power, resulting from synchronized neural activity within a brain region in EEG and MEG recordings, is actually *lower* in the area of cortex where the *attended* stimulus is processed, and *higher* in the area of cortex where the *unattended* stimulus is processed, e.g., [11, 49, 50], although the details are not clear [51]. Moreover, alpha and gamma power seem to be anti-correlated, e.g., [10, 45]. And what about the timing of the alpha power changes? Many of the studies cited show alpha power and coherence results only in a particular small time interval after cue or stimulus onset, typically where power or coherence in the gamma frequency band peaks. In [11], however, alpha power (from MEG recording) in occipital cortex first increased after a top-down directional attention cue (before the target), then decreased sharply around 400 ms after cue onset in cortical areas responsive to both cued and uncued hemifields. Subsequently, alpha power increased more in the occipital cortex ipsilateral to the attended visual field (i.e., the cortex responsive to the unattended visual field) than in the contralateral occipital cortex at 500-700 ms post-cue-onset. This last time interval is when narrow-band transfer entropy, measuring information flow similar to Granger causality, was significant between frontal and parietal areas. More recently, Hanna et al. [50] found similar results to [11]. They used phase transfer entropy with MEG recordings to show that alpha connectivity to, and alpha power in, unattended areas of

sensory cortex (auditory or visual) decreased over time in a coordinated way. This may seem to be a contradiction, because alpha oscillations have been characterized, as just discussed, as inhibitory to sensory processing and to gamma oscillations, and yet in the models studied here (except for CTC) alpha oscillations facilitate information transfer between parts of cortex by interacting with gamma oscillations.

One way to address this apparent contradiction is to consider the alpha power changes recorded by EEG and MEG to be locally-generated (i.e., directly in the indicated cortical area), e.g., [52, 53]. These could be stimulated or signaled in a top-down fashion. Then pulvinar-generated alpha could be the source of the oscillations that organize the gamma oscillations that are associated with good signal transmission [22, 23, 29, 30]. The present model states that phase-offset-matching alpha input would be limited to attended cortex, with alpha input from pulvinar to unattended cortex suppressed via the TRN. Locally-generated alpha, however, could be increased in an unattended sensory area by top-down initiated changes in synaptic efficacies from values generating gamma to values generating alpha, resulting in less gamma, because the same EI pairs can't generate both frequencies. For example, changing the value of a single synaptic efficacy, $S_{IE}$, from 2.1105 in Table 1 to 0.675, changes the oscillation frequency of the quasi-cycles generated by (4), (5) from 40 Hz to 10 Hz. This possibly could happen on a scale of hundreds of milliseconds [54, 55] after a top-down cue, consistent with results regarding timing of responses after cue onset in top-down attention orienting [11, 50]. This would mean it would be more difficult to process stimuli in an unattended area, because active processing is associated with increases in gamma power. Then 10 Hz input from pulvinar could organize the ongoing gamma in the attended area. Alpha originating in the pulvinar would not decrease gamma-generating or gamma power in the attended area because synaptic efficacies wouldn't be changed by input of this alpha. Alpha power would be increased somewhat by the alpha input from the pulvinar (but not to the level of that in unattended cortex), however, but the gamma oscillations would be organized by it, as in [29, 30] and the present study, increasing signal transmission.

Another possible way in which changes in locally-generated alpha and gamma power could be implemented is similar to the experiment described by Chen and colleagues [56], cf. also [57], and modelled by Domhof and Tiesinga [58]. Chen and colleagues found, in mice, that changing the activities of two different types of inhibitory, GABAergic, interneurons dramatically affected the power in lower (5-30 Hz) and higher (20-80 Hz) frequency bands. In particular, inhibiting parvalbumin-expressing (PV) interneurons, which inhibit pyramidal neurons, increased gamma-band oscillations, whereas exciting somatostatin-expressing (SOM) neurons, which inhibit both PV and pyramidal neurons, increased lower-frequency oscillations. This mouse model is quite artificial, being based on optogenetic stimulation of the PV and SOM interneurons, and the switch between frequencies in the wild is usually caused by visual stimulation rather than by top-down signals. The modelling in [58], however, indicates that competition between interneurons can provide a flexible mechanism, perhaps more general than that in the mouse, by which oscillations can be switched among frequency bands on a short time scale. So, a top-down signal/activation potentially could accomplish the increase in local alpha power, while at the same time decreasing gamma power in unattended regions, by changing the activity balance between different types of interneurons.

The predictions of the alpha-gamma model of [29, 30] are captured well by Fig 4E of [29]. In this graph, coherence (amplitude and phase combined from the cross-coherence spectrum) is high between modelled cortical neurons at the alpha frequency in both of two conditions, either a (somewhat noisy) $-\pi/2$ or a $\pi/2$ phase difference between sine wave input. In contrast, coherence is high for gamma frequency only for the $-\pi/2$ alpha phase difference. This is consistent with the Quax et al. finding of a larger signal onset effect for the alpha sine wave input

to the inhibitory cortical model neurons only at the $-\pi/2$ phase difference. This is a result that we explained in our earlier paper [30]. Thus, the model of [29, 30] would predict higher alpha coherence and somewhat high alpha power in cortical regions both in attend-in and in attend-out situations, because alpha coherence and power would not differ depending on alpha phase offset. This does assume that alpha input to all cortical regions from pulvinar is always present, just at different phase offsets depending on whether a stimulus is attended or not. Thus, recordings of pulvinar activity should show alpha input from the pulvinar to the relevant cortical areas but with different phase offsets depending on attend-in versus attend-out conditions. We had attempted to address the problem of determining the optimal phase offset in [30], but on further consideration developed the present model in which optimal alpha input is delivered only to cortical neurons in attend-in situations. Thus, the present model, makes a prediction more similar to the result of Eredath et al. [23]. That is, for the attend-out condition there would be no pulvinar input at all to relevant cortical areas, and thus a lower level of gamma synchronization between cortical areas. In the attend-in condition, in contrast, alpha-band pulvinar input at the optimal phase offset, and thus relatively high gamma synchronization between relevant cortical areas, would be recorded. Of course, in all cases pulvinar alpha input, although noisy, would be fairly highly synchronized because alpha input would be generated with a constant phase offset.

A problem with these predictions arises with respect to what would be observed in cortex. The above analysis of the interaction of local changes in alpha-band activity with alpha input delivered from the pulvinar implies a complicated picture. This picture also changes over a time span of hundreds of milliseconds, as local synaptic or interneuron balance changes, and pulvinar activity responds to TRN input and thus changes its input to cortex. Thus, recordings of both cortical and pulvinar activity and coherence over relatively long time spans as a function of attention differences are needed to adequately test the alpha-gamma models. The snapshots provided by experiments to date reveal tantalizing hints of what could be observed, but do not cleanly disconfirm or confirm any of the mentioned models.

Another interesting prediction of the alpha-gamma model is encompassed in the signal onset effect. In this model, as shown in our simulations, when alpha and gamma phase offsets roughly match there is an transient increase in oscillation amplitude at signal onset, followed by a decrease in amplitude to an above baseline level as long as the signal lasts (Fig 1. This effect is hypothesized to contribute to the increased effectiveness of the attended signal. We know of no experimental study focussed on detecting this hypothesized effect, although that of Fries et al. [45] does report an increase in gamma power (attend-in relative to attend-out) in V4 shortly after signal onset, that then declines to an above baseline level. In the alpha-gamma model this should happen to the composite alpha+gamma oscillation. This would be a useful test of the alpha-gamma model, confirming it if present and disconfirming it if absent. A problem would be disentangling it from the typical increase in neural firing rate, and subsequent neural adaptation, upon signal onset, which has been widely studied, e.g., [59], although it is consistent with that phenomenon.

Another issue is the relationship of the present results, and those of previous models of [29, 30], to the work on cross-frequency coupling (CFC), which usually takes the form of phase-amplitude coupling (PAC) (reviewed by Esghaei et al. [60]). Here we focus on PAC. PAC involves the amplitude of a higher frequency process, which could be bursts of spikes, spectral power, firing rate, etc., being 'locked' (stochastically) to the phase of a lower-frequency oscillation, either low (30-80 Hz) or high (80-200 Hz) gamma power locked to alpha or theta phase. Typically, larger firing bursts, more spectral power, etc. at the higher frequency occur around a particular phase, usually at a peak or a trough, of the lower frequency oscillation. PAC is linked to performance during a variety of perceptual and cognitive tasks and has been reported in

both human (e.g., [10, 61, 62] and animal (e.g., [63, 64]) studies, although the role of PAC seems to differ depending on task and brain regions involved. Moreover, PAC can occur both within and between brain regions. In particular, the evidence for the involvement of PAC in attention focussing is conflicting, with some studies reporting a decrease in PAC within attend-in areas (e.g., [63]), whereas others report an increase (e.g., [65]). It is possible that PAC plays different roles in peripheral and more central attention modulators, as hypothesized by Esghaei et al. [60]. Importantly, the alpha-gamma models we study here all imply that alpha-gamma PAC would occur as a result of pulvinar alpha input to the cortex. Figs 3E in [29] and 3A and 4A of [30] clearly demonstrate coupling of gamma power and gamma amplitude, respectively, to the phase of the input alpha oscillation. Given the complications described above in the role of alpha oscillations in attention focussing, however, it is difficult to predict exactly what would be seen in the ideal experiment. Thus, although PAC seems to play a role in attention focusing, possibly along the lines suggested by Esghaei et al. [60], it is not yet clear what that role encompasses.

## Conclusions

We have elaborated a model of attentional selection in which alpha oscillations generated in the pulvinar nucleus of the thalamus organize gamma oscillations in cortical areas so that they are more coherent than without the alpha input. Thus, they transfer information about signals more efficiently than without that input. This model describes how an attention system can select a particular stimulus to emphasize, either through a top-down signal from other cortical areas, or by preferring more salient stimuli in a bottom-up fashion. There are significant limitations to the model as it exists, however. In particular, the model works well only in a limited parameter range, so it would be very sensitive to physiological conditions. Moreover, there is no consensus on how to characterize stimuli (or signals) in such models. Some researchers simply add a constant to the input, as is often done in physiological studies in vitro. Others, including us, have argued that signals should be inputs that increase the amplitude of the LFP, for example outputs of leaky integrate and fire neurons [42].

The relation between coherence and communication is not as simple as it has seemed from the CTC hypothesis. First, through input from a third area (e.g., the pulvinar), coherence can be increased even though there is no direct communication at all between the monitored areas. This can fool data analysis that only focuses on, e.g., phase coherence between pairs of cortical areas. Second, coherence between cortical areas can be increased simply by one area sending spikes to another: coherence through communication (CTC′). Third, it is not clear how to measure communication in these models or in experimental data. Some studies measure information transmission by focusing on the onset of the stimuli, whereas others focus on the entire time the stimulus is present, with somewhat different results.

Finally, it is becoming clear that alpha oscillations could perform many roles in the brain, from attention selection to timing to long-range coordination. It is important that these roles and their individual circumstances be clarified. Future modelling and empirical work might be able to determine when and how the different suggested mechanisms perform. But it will be important to remember the scope and limitations of these mechanisms. It is also important to determine whether the ubiquitous neural oscillations play a dynamic and/or causal role in any of these mechanisms, or simply arise epiphenomonally from the spiking of neurons as they transmit information throughout the brain.

## Author Contributions

**Conceptualization:** Priscilla E. Greenwood, Lawrence M. Ward.

**Data curation:** Lawrence M. Ward.

**Formal analysis:** Priscilla E. Greenwood, Lawrence M. Ward.

**Funding acquisition:** Lawrence M. Ward.

**Investigation:** Priscilla E. Greenwood, Lawrence M. Ward.

**Methodology:** Lawrence M. Ward.

**Project administration:** Lawrence M. Ward.

**Resources:** Lawrence M. Ward.

**Software:** Lawrence M. Ward.

**Supervision:** Lawrence M. Ward.

**Validation:** Lawrence M. Ward.

**Visualization:** Lawrence M. Ward.

**Writing – original draft:** Priscilla E. Greenwood, Lawrence M. Ward.

**Writing – review & editing:** Priscilla E. Greenwood, Lawrence M. Ward.

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
