## [Decision Letter · Decision Letter 0]

13 Feb 2024

Dear Dr. Ward,

Thank you very much for submitting your manuscript "Attentional selection and communication through coherence: Scope and limitations." for consideration at PLOS Computational Biology.

As with all papers reviewed by the journal, your manuscript was reviewed by members of the editorial board and by several independent reviewers. In light of the reviews (below this email), we would like to invite the resubmission of a significantly-revised version that takes into account the reviewers' comments.

As one major point, both reviewers asked to better discuss existing literature, and to also report experimental results which contradict model results/predictions. Limitations of the model should be stated clearly, and claims about the explanatory power of the modelling approach reduced accordingly.

We cannot make any decision about publication until we have seen the revised manuscript and your response to the reviewers' comments. Your revised manuscript is also likely to be sent to reviewers for further evaluation.

Sincerely,

Udo A. Ernst

Guest Editor

PLOS Computational Biology

Thomas Serre

Section Editor

PLOS Computational Biology

Reviewer's Responses to Questions

**Comments to the Authors:**

Reviewer #1: The review is uploaded as an attachment.

Reviewer #2: In this work, the authors expand on a previous model that described how the interactions of two "cortical" populations oscillating at 40Hz are coordinated by oscillations of two "pulvinar" populations oscillating at 10Hz. The previous work established the "best" phase offset between the two gamma and alpha oscillations for signal transmission. In this work, they extend this to two separate cortical populations in each area to explain selective routing of information as is seen in the context of attentional selection. They discuss the benefits and limitations of this model. It is well-written and has a rather broad scope.

The work is generally solid and rigorously tests a simple model of phase-dependent routing of information during "attention". It nicely captures several experimentally observed effects such as a boosting of coherence and signal transmission for an "attended" population that can be selected either in a top-down or bottom-up process. They also make explicit that high coherence values between two areas do not necessarily mean that information exchange is happening.

However, there are also several drawbacks, discussed below:

In general the paper would benefit from more graphical representations of the data. Except for the model schematic and the figure from a previous paper, there is only a table for all results. A graphical representation of how the signal (enhancement), coherence and MI is affected by phase difference, connectivity strength, noise,... would be beneficial for the reader to understand the relationships between these different variables and the different effect sizes and compare them. Similarly, the way the "signal onset effect" (ln 264), "signal onset enhancement" (ln 283) and "signal enhancement effect" (ln 284) (are they all the same?) is computed, is not entirely clear. Here, too, a graphical representation of what max and mean measures mean would be helpful. They should plot an example, perhaps the one from row 1 of their example table, and visualize the actual signals (V1 and V2), the signal time course and their readout of the max and mean signal effects.

Another problem is, that the work here is not really well understood without knowledge of the previous paper. They do briefly introduce this model, but then immediately jump to the discussion of Figure 1 and the signal onset effects, which have not been introduced to the reader. Figure 1 is not really understandable without reading the previous paper. They should explain their measures and hypotheses better for the reader. If their point for Figure 1 is, that both quasi cycles and noisy sine-waves for the alpha yield similar results, they should also choose the same phase offset. If their point is, that the signal propagation is maximal, when the phase offsets match, one part of the figure would likely also be enough. Instead, a figure of the respective oscillations with the added signal and how it propagates to the other area, for example for two exemplar phase offsets could be more helpful.

A question that might be discussed would be, that in this model, top-down attentional selection works by inhibiting the Pulvinar population that corresponds to the unattended signal. However, in reality, there is not just one other population, but often the entire visual field, except the attended location (for spatial attention) or all neurons NOT encoding the attended feature (for feature-based attention). This would mean the higher level attentional process would need to drive all but the attentionally selected reticular populations. How could this be biologically plausibly implemented? Especially in the case of feature-based attention (without a retinotopic map) this might be difficult to achieve.

It is unclear from the text, whether the capture of attention by a salient signal would be enough to win the TRN competition alone, or whether the reallocation of top-down bias input is additionally strictly necessary. In all cases the authors discuss bottom-up attentional capture, they also mention the reallocation of top-down attention. But could the attention be shifted also without this? How big would the signal differences need to be to win the TRN competition? The need to invoke top-down processes for all attentional shifts would seem to be a rather slow mechanism, involving the traveling of the new salient signal to frontal areas and back, before a saccade could happen to the new attended location. But perhaps this is not necessary. However, this is not clear from the manuscript.

A precarious point about the plausibility of this model for attentional effects is the enormous signal-boosting effect the authors see when adding noise to the signal, even without any additional alpha inputs. Here, instead of a 10% or so boost they see with "attention", they suddenly see a more than tripling of the signal amplitude, simply by adding noise without attending. They mention this, but don't truly discuss why this large mismatch in effect size comes about, what it could mean, and/or how the level of noise relates to this.

Regarding Coherence through Connectivity: In Schneider et al. the main point is, that coherence is seen in the LFP simply through rhythmic synaptic inputs to a target area, even without an ongoing oscillation or spike-entrainment in this target area. This case is however never tested in this model, as, even without an alpha input, all cortical areas always still oscillate at the same gamma frequency, putatively responsible for a lot of the coherence they see. The discussion of this in the results is rather short compared to how much the authors highlight this point in the introduction and also the abstract. Their argumentation of this putatively being the mechanism for "unattended" information transfer needs further clarification for the reader.

The section "Onset vs total signal effects" is raising a perhaps interesting point. Yet the reader cannot judge how transient the signal-boosting effect is, without ever seeing a time-course. I reiterate my point of more graphical representations being extremely useful for the reader.

For the section "Paradox re alpha", the authors should either include this in the discussion or perform actual simulations with ongoing, competing alpha and gamma oscillations in the two cortices as they suggest here. This would be testable. Otherwise, this should not be in the results section because it is purely speculative. Biophysically it seems highly unlikely, that the gamma oscillatory (likely a PING mechanism) could on a millisecond time scale (as they discuss here) switch to suddenly be alpha oscillators by changes in synaptic properties. Similarly, their discussion of the Chen results seems misguided. There is no evidence in the field that SOM cells are involved in the generation of alpha oscillations. A highly unnatural optogenetic drive to SOM cells at 5Hz makes the circuit resonate at that frequency, however, this is not plausibly happening under naturalistic conditions. SOM cells have instead been shown to be responsible for a low gamma oscillation between about 25 and 30Hz by Chen et al. and others. Also, these experiments were done in mice, which do not show alpha oscillations with properties comparable to primates.

**Have the authors made all data and (if applicable) computational code underlying the findings in their manuscript fully available?**

Reviewer #1: Yes

Reviewer #2: Yes

PLOS authors have the option to publish the peer review history of their article (what does this mean?). If published, this will include your full peer review and any attached files.

Reviewer #1: No

Reviewer #2: No
---

## [Decision Letter · Decision Letter 1]

25 Jun 2024

Dear Dr. Ward,

Thank you very much for submitting your manuscript "Attentional selection and communication through coherence: Scope and limitations." for consideration at PLOS Computational Biology. As with all papers reviewed by the journal, your manuscript was reviewed by members of the editorial board and by several independent reviewers. The reviewers appreciated the attention to an important topic. Based on the reviews, we are likely to accept this manuscript for publication, providing that you modify the manuscript according to the remaining recommendations of the second reviewer.. .

Sincerely,

Udo A. Ernst

Guest Editor

PLOS Computational Biology

Thomas Serre

Section Editor

PLOS Computational Biology

Reviewer's Responses to Questions

**Comments to the Authors:**

Reviewer #1: Dear Authors,

you have addressed all my previous concerns comprehensively. The inclusion of the missing arguments has strengthened the discussion significantly.

From my side, there are no further objections, and I am happy to recommend the manuscript for publication.

Reviewer #2: This manuscript has significantly improved. The starting point is explained much better and should now also be understandable without having to read the previous paper. It discusses the possible mechanism for attentional switching, where letting pulvinar alpha arrange gamma oscillations between V1 and V2 leads to enhanced signal transmission. It also outlines how and where the model fails to capture experimentally observed attentional effects or acts erratically. The discussion has been significantly expanded, adding more nuance to the results and placing them into context much better.

The graphical depictions of the signals and the onset effects are extremely helpful for the reader. Where they depict the signal time courses (Figures 1 and 4), they could, however, label the x-axes with seconds or ms, not with samples as they have done here. That way, it would also be straightforward to see whether the oscillation that is visible is alpha or gamma, which is now only possible by checking the methods.

However, I reiterate my point of potentially also plotting different outputs of the model against the input variables to get a better idea of their relationships. Mean sigoe as a function of signal strength, signal noise, connectivity strength... Many of these are discussed in writing, but a simple plot is much more directly conveying this information to the reader than extracting this information from a table or writing.

Is there any experimental evidence, that the large, transient increase in oscillatory strength at the onset of the signal also happens in experiments?

minor:

line 93: "maximum" should be "maximal"

sentence in ln 101 ff could be rephrased for clarity

ln 105 "our" should be "their" to keep it third person as they started.

ln 737: "though" should be "through"

**Have the authors made all data and (if applicable) computational code underlying the findings in their manuscript fully available?**

Reviewer #1: Yes

Reviewer #2: **No: **

PLOS authors have the option to publish the peer review history of their article (what does this mean?). If published, this will include your full peer review and any attached files.

Reviewer #1: No

Reviewer #2: No

Figure Files:

Data Requirements:

Reproducibility:

References:

---

## [Editor Report · Decision Letter 2]

22 Jul 2024

Dear Dr. Ward,

We are pleased to inform you that your manuscript 'Attentional selection and communication through coherence: Scope and limitations.' has been provisionally accepted for publication in PLOS Computational Biology.

Best regards,

Udo A. Ernst

Guest Editor

PLOS Computational Biology

Thomas Serre

Section Editor

PLOS Computational Biology

---

## [Editor Report · Acceptance letter]

30 Jul 2024

PCOMPBIOL-D-23-01312R2 

Attentional selection and communication through coherence: Scope and limitations.

Dear Dr Ward,

I am pleased to inform you that your manuscript has been formally accepted for publication in PLOS Computational Biology. Your manuscript is now with our production department and you will be notified of the publication date in due course.

With kind regards,

Anita Estes
